

# Identifying optical microscope images of CVD-grown two-dimensional MoS$_2$ by convolutional neural networks and transfer learning

Cahit Perkgoz

Department of Computer Engineering, Eskisehir Technical University, Eskişehir, Turkey

## ABSTRACT

**Background**. In Complementary Metal-Oxide Semiconductor (CMOS) technology, scaling down has been a key strategy to improve chip performance and reduce power losses. However, challenges such as sub-threshold leakage and gate leakage, resulting from short-channel effects, contribute to an increase in distributed static power. Two-dimensional transition metal dichalcogenides (2D TMDs) emerge as potential solutions, serving as channel materials with steep sub-threshold swings and lower power consumption. However, the production and development of these 2-dimensional materials require some time-consuming tasks. In order to employ them in different fields, including chip technology, it is crucial to ensure that their production meets the required standards of quality and uniformity; in this context, deep learning techniques show significant potential.

**Methods**. This research introduces a transfer learning-based deep convolutional neural network (CNN) to classify chemical vapor deposition (CVD) grown molybdenum disulfide (MoS$_2$) flakes based on their uniformity or the occurrence of defects affecting electronic properties. Acquiring and labeling a sufficient number of microscope images for CNN training may not be realistic. To address this challenge, artificial images were generated using Fresnel equations to pre-train the CNN. Subsequently, accuracy was improved through fine-tuning with a limited set of real images.

**Results**. The proposed transfer learning-based CNN method significantly improved all measurement metrics with respect to the ordinary CNNs. The initial CNN, trained with limited data and without transfer learning, achieved 68% average accuracy for binary classification. Through transfer learning and artificial images, the same CNN achieved 85% average accuracy, demonstrating an average increase of approximately 17%. While this study specifically focuses on MoS$_2$ structures, the same methodology can be extended to other 2-dimensional materials by simply incorporating their specific parameters when generating artificial images.

Corresponding author
Cahit Perkgoz,
cahitperkgoz@eskisehir.edu.tr

## INTRODUCTION

In recent years, there has been a growing focus on two-dimensional (2D) materials, spurred by the seminal discovery of graphene (*Geim & Novoselov, 2007*) and resulting in a prominent area of research (*Zhang et al., 2019c*). Ongoing investigations have unveiled the exceptional and superior properties inherent to these 2D materials, leading to significant progress in their characterization and practical implementation (*Novoselov et al., 2005*). Transition metal dichalcogenides (TMDs) have garnered substantial interest from researchers in both materials science and device applications, owing to their distinctive optoelectronic attributes, including a direct bandgap in their monolayer form (*Geim & Novoselov, 2007*; *Zhang, 2018*). Moreover, 2D TMDs exhibit considerable potential as channel materials in metal-oxide-semiconductor field-effect transistors (MOSFETs) due to their noteworthy ability to reduce off-state current and scale dimensions, addressing a critical aspect of power dissipation in future electronic chips (*Hua et al., 2020*).

To overcome the limitations of gapless graphene for future switching devices, monolayer TMDs offer a solution with their relatively large bandgap. Among TMDs, molybdenum disulfide ($MoS_2$) has been one of the most extensively studied materials (*Perea-López et al., 2014*). It comprises covalently bonded Mo and S atoms, forming layers held together by weak van der Waals forces (*Fortin & Sears, 1982*; *Zhang, Wan & Yang, 2019b*).

Transitioning these 2D materials from lab-scale studies to device implementation and mass production poses a challenge in achieving large-scale and controlled growth. While initial 2D materials were obtained through mechanical exfoliation (*Yi & Shen, 2015*), other methods (*Bonaccorso et al., 2012*) have also been proposed. Presently, chemical vapor deposition (CVD) stands out as the most promising technique for producing large and uniform 2D films or flakes with desired properties (*Liu, Wong & Chi, 2015*), given its compatibility with microelectronic processes (*Liu et al., 2019*). This method enables the growth of high-quality, larger-area, and single-layer TMDs, including $MoS_2$ flakes (*Perkgoz & Bay, 2016*). Nevertheless, scalability and wafer-size industrial fabrication persist as challenges, along with other issues such as contact resistance, doping, and charge transfer problems (*Lin et al., 2016*). Therefore, considering various perspectives and objectives, it is imperative to cultivate 2D TMDs on diverse substrates and conduct their characterization in a practical and expeditious manner for the development of various device architectures.

In general, the initial and cheapest characterization tool used for identifying 2D materials is an optical microscope, facilitating the easy capture of images of the synthesized films and flakes (*Yorulmaz et al., 2019*). Additionally, various characterization techniques, including Raman scattering spectroscopy, atomic force microscopy (AFM), and photoluminescence (PL) spectroscopy, are employed to reveal the structural, vibrational, and optical properties of these materials (*Özden et al., 2016*; *Zhang et al., 2019a*). However, in order to achieve ambitious goals in 2D materials research, such as attaining low-energy electronics (*Pal et al., 2021*) and integrated circuits utilizing atomically-thin or 2D van der Waals materials (*Jiang et al., 2019*), it becomes imperative to cultivate large-area and uniformly grown 2D materials. Consequently, a thorough analysis of the deposited 2D structures across the entire substrate surface is essential. On the other hand, conventional characterization techniques

like AFM and Raman spectroscopy become cumbersome and time-consuming when applied to such expansive areas. In this context, the utilization of an optical microscope for 2D material identification proves immensely advantageous due to its simplicity and speed.

From another perspective, manually distinguishing normal or defective structures can be both challenging and time-consuming, and naturally the effectiveness of the optical microscope will depend on the experience of the operator. For these reasons, researchers have increasingly shifted their focus in recent years towards the development of deep learning-based methods for detection processes (*Al-Waisy et al., 2022*; *Lin et al., 2018*; *Ngome Okello et al., 2021*; *Zhang et al., 2023*). In a general sense, deep learning constitutes a subset of machine learning based on artificial neural networks, encompassing specialized architectures designed for deep learning (*Saito et al., 2019*). Presently, such methods find application in diverse fields such as biology, physics, medicine, and electronics (*Dong et al., 2023*; *Masubuchi & Machida, 2019*; *Shinde & Shah, 2018*; *Shorten, Khoshgoftaar & Furht, 2021*). The acceleration and automation of scientific discoveries through the utilization of deep learning techniques have been a subject of research and practical implementation for numerous years. Furthermore, deep learning methods prove valuable for tasks like image processing, image classification, image interpretation, and image generation (*Masubuchi et al., 2020*). Notably, one of the most prominent algorithms within the realm of deep learning is deep convolutional neural networks (CNN) (*Alzubaidi et al., 2021*; *Yao, Lei & Zhong, 2019*), which has demonstrated remarkable success and performance in image recognition and classification. Consequently, it is frequently favored over alternative approaches in the realm of deep learning, especially when dealing with image-related tasks (*Bhuvaneswari et al., 2021*; *Shi et al., 2023*).

In order to effectively use CNNs, acquiring a sufficient number of images is essential. However, as previously mentioned, generating images of CVD-grown 2D material structures, and labeling the data is a labor-intensive and demanding process, resulting in limited training data for image classification. Transfer learning offers a strategy to acquire additional data and knowledge from related domains and apply it to the specific problem (*Bozinovski, 2020*; *Bozinovski & Fulgosi, 1976*; *Tan et al., 2018*; *West, Ventura & Warnick, 2007*). However, due to the unique nature of the problem in this study, it is more appropriate to generate realistic artificial images using Fresnel equations. These artificially generated images are then used for pre-training, effectively alleviating the data constraint issue for CNN algorithms. Subsequently, fine-tuning is performed using real images obtained *via* optical microscopy. In summary, the aim of this article is to utilize a transfer learning-based CNN method to automatically classify 2-dimensional MoS$_2$ structures, obtained by CVD technique, as normal or defective. Although there are a few studies on the analysis of 2D materials using deep learning, they either aim to estimate the thickness of the 2D materials (*Lin et al., 2018*) or determine the defects on atomic-level images by scanning transmission electron microscopy analysis, which is a very expensive and time-consuming process (*Ngome Okello et al., 2021*). In the literature, a deep learning approach that analyzes normal or defective 2D MoS$_2$ materials obtained by CVD and utilizes Fresnel equations to generate data for the transfer learning method has not been encountered.

While numerous research groups have investigated 2D materials using exfoliation techniques, it is evident that these exfoliated materials are not optimally suited for the design of electronic devices, particularly chip technology due to their small size and low yield in production. As a result, many research laboratories have turned to chemical vapor deposition (CVD) techniques to obtain such 2D materials also because of their compatibility in microelectronics. However, despite the high potential of CVD for achieving such innovative materials, the obtained structures are not fully uniform, and their analysis is critical for their wide usage in different applications. It will be highly beneficial to simplify this time-consuming task of 2D material analysis, often requiring specialized expertise. Hence, there is a research gap that lies in the methods capable of expediting the analysis of materials, specifically 2D ones, produced through CVD. Up to present, deep learning methods have only been utilized in exfoliated flakes or ALD grown structures. Addressing this gap, the present study contributes significantly to two key areas: (1) Artificial two-dimensional $MoS_2$ images are efficiently generated using Fresnel equations, offering a novel approach to data augmentation; and (2) the study proposes a two-stage convolutional neural network (CNN) that employs transfer learning to enhance classification accuracy, especially in scenarios with limited datasets, which is the case in experimental research laboratories. These contributions aim to automatize and advance the analysis of two-dimensional materials, which are among the promising alternatives that can replace silicon-based chip technology.

This article unfolds in the following manner: 'Materials & Methods' delves into the methodology for creating artificial images and introduces the concept of CNN. 'Results and Discussions' covers the experiments and their respective results, while 'Conclusions' provides concluding remarks.

## MATERIALS & METHODS

The primary objective of this research is to employ artificial intelligence in determining the suitability of 2D structures grown through CVD technique for electronic device applications. Traditionally, the initial step involves the manual analysis of the produced 2D structures using an optical microscope, representing one of the most labor-intensive methods available. While other methods do exist, they tend to be expensive and may not be practical for analyzing large areas. In contrast, the proposed approach leverages computational tools, specifically convolutional neural networks, to automatically characterize and label these images based on defects or irregularly shaped flakes, thereby assessing their viability for device fabrication. However, training a CNN is contingent on the availability of a substantial amount of data, which may prove challenging to collect and label manually for each sample. Furthermore, in many research centers, acquiring sufficient data remains a hurdle. To augment the dataset, Fresnel equations, tailored to the specific optical properties of the aforementioned materials, can be employed to generate artificial images of $MoS_2$ flakes on a substrate ($SiO_2/Si$), which can then be utilized in the transfer learning process. This section will begin by providing an explanation of transfer learning, followed by a discussion on the application of Fresnel equations to produce synthetic images. Finally, the concept and architecture of the CNN will be presented.

## Transfer learning

While machine learning methods exhibit remarkable capabilities in solving complex problems, their effectiveness hinges on the availability of sufficient amount of training data. Challenges in data acquisition may arise due to factors such as high costs, time constraints, and confidentiality concerns. In cases where acquiring specific data is challenging, data obtained from any analogous system, device, or process, related to the existing problem in need of a solution can be used in deep learning as a preliminary training process before training with data specific to the problem (*Tan et al., 2018*). For instance, addressing an issue with a spacecraft's engine might require more data than is currently available. In such cases, data from aircraft engines, which share similarities, can be employed in the pre-learning task. Similarly, the characterization of 2D materials faces a parallel challenge, and obtaining a sufficient number of images is a demanding process. Additionally, analyzing enlarged $MoS_2$ flakes for classification demands significant time from researchers due to the high pixel count even in a single image. However, finding analogous images for pre-training in the transfer learning method may not be feasible due to the uncommon nature of the problem in this study. On the other hand, leveraging Fresnel equations to artificially generate images offers a more precise and reliable means to mitigate this dependency. Data generation can be swiftly accomplished within seconds with the aid of an algorithm capable of generating realistic 2D material images using mathematical expressions. Subsequently, these artificially generated data serve as the foundation for pre-training of the CNN algorithm. Then, the limited real optical microscope images can be used to fine-tune the parameters in the CNN. In this study, the Fresnel equations, elaborated upon in the subsequent section, were initially employed to generate artificial images based on the reflection's light intensity values from various regions within an image, which were then utilized in the pre-training phase.

## Fresnel equations

In a brightfield optical microscope image, structures become visible due to the contrast in reflected light from various regions on the examined samples. When examining 2D materials with this type of microscope, they become visible due to the contrast (Eq. (1)) between the material and the substrate on which they are grown (Fig. 1), enabling their analysis.

As shown in Fig. 1, there are two distinct regions, with region A containing an additional $MoS_2$ layer on $SiO_2$ and Si layers. For the sake of readability, the layers of air, $MoS_2$, $SiO_2$ and Si are indexed as 0, 1, 2 and 3, respectively. A $SiO_2$ layer of 300 nm and Si are commonly chosen as a substrate due the high contrast observed with the 2D material (region A) compared to the absence of the material (region B) (*Ciregan, Meier & Schmidhuber, 2012*). The contrast ($C$) for these regions is as follows:

$$C = \frac{I_A - I_B}{I_A + I_B} \tag{1}$$

where $I_A$ and $I_B$ are the light intensities in region A and region B, respectively. The intensities can be expressed as in Eqs. (2) and (3).

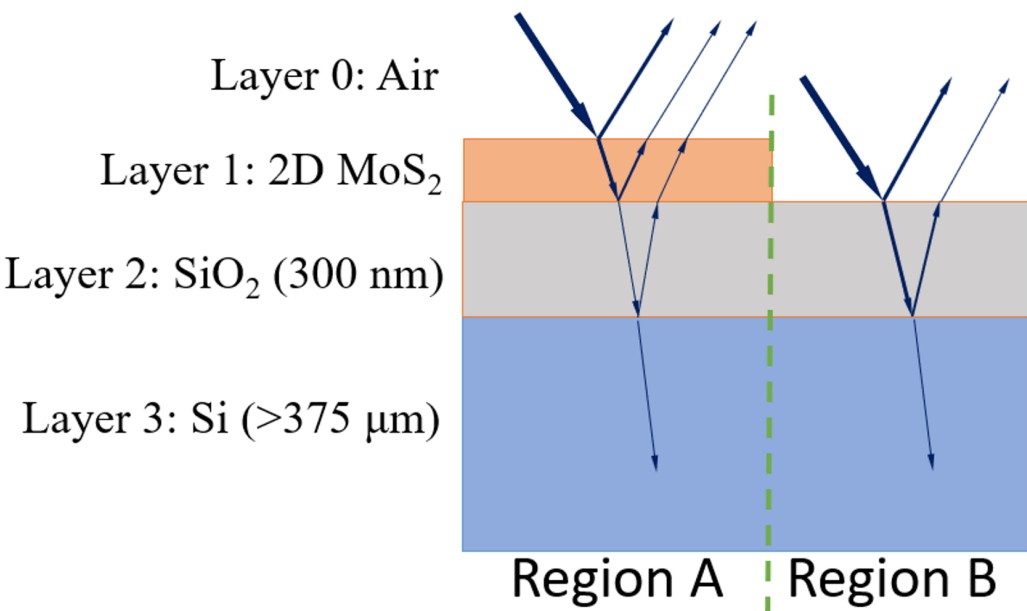

**Figure 1  Cross-section of a grown 2D material (MoS₂) on a substrate (SiO₂/Si).**

$$I_A = |\bar{r}_A r_A| \tag{2}$$

$$I_B = |\bar{r}_B r_B| \tag{3}$$

where, $r_A$ and $r_B$ are the reflection Fresnel coefficients for the related regions and can be calculated using the refractive indices and phase shifts (*Blake et al., 2007*; *Zhang et al., 2021*), as given in Eqs. (4) and (5).

$$r_A = \frac{r_{01}e^{i\alpha} + r_{12}e^{-i\beta} + r_{23}e^{-i\alpha} + r_{01}r_{12}r_{23}e^{i\beta}}{e^{i\alpha} + r_{01}r_{12}e^{-i\beta} + r_{01}r_{23}e^{-i\alpha} + r_{12}r_{23}e^{i\beta}} \tag{4}$$

$$r_B = \frac{r_{01} - r_{12}e^{-i2\Phi_1}}{1 + r_{01}r_{12}e^{-i2\Phi_1}} \tag{5}$$

where $\alpha = \Phi_1 + \Phi_2$ and $\beta = \Phi_1 - \Phi_2$.

Fresnel coefficients ($r_{ij}$) are functions of refractive indices ($n_i$ and $n_j$), where layer $i$ is above layer $j$, as given in Eq. (6). Specifically, there are three coefficients: $r_{01}$, $r_{12}$, and $r_{23}$. These coefficients represent the relative refractive indices between air and MoS₂, MoS₂ and SiO₂, and SiO₂ and Si, respectively.

$$r_{ij} = \frac{(n_i - n_j)}{(n_i + n_j)} \tag{6}$$

Due to the varying thickness of different layers, a phase shift $\Phi_i$ occurs in each layer during the propagation of light in each medium. This phase shift is defined by Eq. (7):

$$\Phi_i = \frac{2\pi\, n_i d_i}{\lambda} \tag{7}$$

where $n_i$, $d_i$ and $\lambda$ represent the refractive index, the thickness of the medium $i$, and the wavelength of the light, respectively.

As a result, when a stack of layers made of different materials is observed under brightfield optical microscope lighting, varying intensities will be obtained, as illustrated in Fig. 1. The upper layers of region A and B consist of 2D material ($MoS_2$) and $SiO_2$, respectively. The thickness of $SiO_2$ is chosen to be 300 nm, providing high contrast, and used as the standard thickness in the growth experiments (*Blake et al., 2007*). The light propagates through different materials, taking paths of different lengths, which results in a phase shift. Consequently, regions A and B will exhibit different total reflection intensity values as defined in Eqs. (4) and (5). The red, green, and blue channels of an RGB image are obtained by using the wavelength corresponding to these colors. These RGB intensity values can then be used to construct the artificial images of a sample, analogous to real images produced experimentally.

In optical microscopy, tungsten-filament light bulbs, white LEDs, xenon, and mercury lamps can be used as white light sources for illuminating the specimen. These artificial sources combine appropriate amounts of red, blue, and green light to generate white light. While the spectral distribution curves vary for each of these sources, a common characteristic among them is the broad spectral range, particularly in the green and red parts (*Tawfik, Tonnellier & Sansom, 2018*). Therefore, to achieve this effect and align with the typical spectra of various white light sources, the intensities of red, green, and blue are adjusted accordingly. Various color filters can be used to make images more visible in microscopes. During the pre-training phase, generating images by manipulating intensity across different wavelengths, as opposed to images produced within a specific spectrum of light, contributes to increased diversity in the dataset. This, in turn, mitigates overfitting during the pre-learning phase by preventing algorithms from memorizing encountered instances, thereby addressing overfitting challenges in AI algorithms.

## Convolutional neural networks

The convolutional neural network is a specialized type of artificial neural network designed specifically for computer vision tasks, such as image classification and object detection. Early studies on CNNs originated from the discovery of simple, complex, and hyper-complex cells in the visual cortex of animals, which play a role in recognizing objects through their eyes (*Hubel & Wiesel, 1959*; *Hubel & Wiesel, 1968*). Based on these studies, the *neocognitron* was introduced, describing the layers of a CNN to imitate such complex and simple cells (*Fukushima, 1980*). To identify patterns, CNN employs mathematical operations from linear algebra, particularly matrix multiplications, demanding substantial computational power. This computational complexity initially hindered the immediate adoption of CNNs and other neural network techniques following their invention. However, advancements in computer technology have paved the way for CNNs to address various challenges

through diverse architectures in numerous fields (*Alzubaidi et al., 2021*), including image processing (*Schmidhuber, Meier & Ciresan, 2012*), video analysis (*Ji et al., 2012*), medical image analysis (*Tajbakhsh et al., 2016*), natural language processing (*Collobert & Weston, 2008*), and time series analysis (*Tsantekidis et al., 2017*). Compared to other neural network techniques, CNNs involve fewer computations, as they utilize a shared set of weights (filters or kernels) for different regions within an input matrix, such as images. These filters slide over the input matrix, performing convolution operations to extract features.

Essentially, a CNN architecture comprises a convolution layer, a pooling layer, and a fully connected layer, in addition to the input and output layers (Fig. 2). The distinctive properties of a CNN, distinguishing it from other types of neural networks, are the inclusion of a convolution layer responsible for feature extraction from an image and a pooling layer that reduces the size of the input following the convolution operation. These layers can be applied iteratively, capturing low-level features in earlier layers and high-level features in subsequent ones, with the information extracted at each level progressing through the layers.

A CNN is trained using a labeled dataset. In each iteration, forward operations are conducted through the layers, resulting in an output vector. Subsequently, an error is calculated using the desired outputs provided with the dataset. Optimization techniques are then applied to adjust the parameters (filter weights, biases) of the network through backward operations. This process aims to reduce the total error at the end of the subsequent forward operations. The training concludes when the error falls below a predetermined level. At this point, the network possesses optimal parameters and can be tested on unseen data to evaluate its success. The functions of typical layers of a CNN are explained in the following.

### Input layer

The typical input for a CNN is an image, where the values of this image are organized into a matrix, creating the input layer. An RGB image, for instance, can be represented as a matrix with dimensions $M \times N \times 3$, where $M$ is the height, $N$ is the width of the image, and 3 represents the depth for the red, green, and blue channels. Normalizing the elements of the input matrix at the beginning of the algorithm facilitates the rapid discovery of optimized parameters during training.

### Convolution layer

The key layer in a CNN structure is the convolution layer, responsible for performing the convolution operation on the input matrix using a set number of filters. Each filter is designed to detect a distinct feature, mapping it to the next layer. This process entails sliding the filter across the image and applying it to different areas, essentially performing a dot product of the receptive field and the filter at each step. The step size (stride) can be any positive integer. For an $M \times N \times d$ size image, an $m \times n \times d$ size filter, and a stride of 1, the convolution layer's output will generate a new matrix with dimensions $(M-m+1) \times (N-n+1)$. The algorithm may involve the application of multiple filters, and if $k$ number of filters are applied to the image, the output of this convolution layer will be a

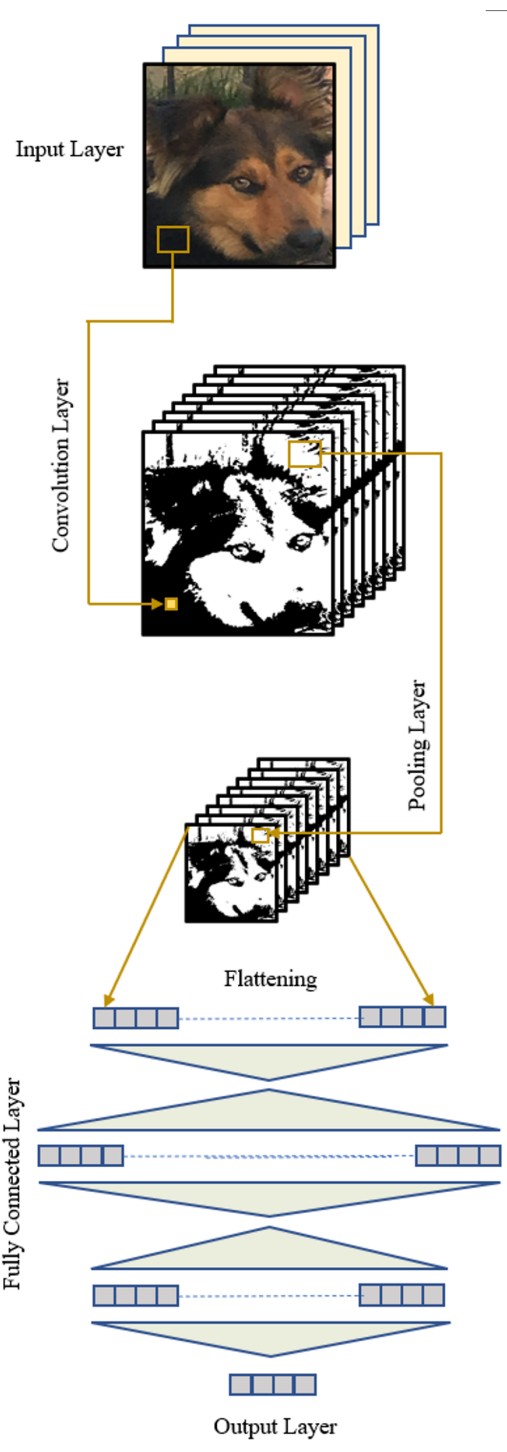

**Figure 2  A CNN structure.**

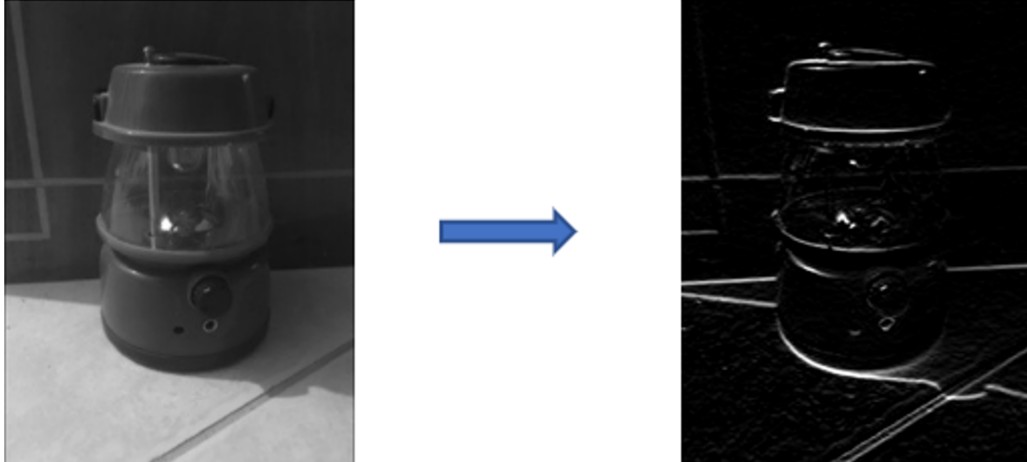

**Figure 3  Input and output of a convolution operation with Sobel filter.**

tensor of size $(M - m + 1) \times (N - n + 1) \times k$. The convolution equation is provided below:

$$C_{i,j}^k = \sum_{c=1}^{d}\sum_{b=1}^{n}\sum_{a=1}^{m} I_{(i+a-1),(j+b-1),c} f_{a,b,c}^k \tag{8}$$

where $C$ denotes the feature map, which is the outcome of the convolution between $I$ and $f$. $I$ is the input matrix, $f$ is the filter matrix, $m$ is the height and $n$ is the width of the filter. $d$ is the depth of both the input matrix and filter. As an example, the outcome of convolving an image with a Sobel filter, designed to detect horizontal edges, is illustrated in Fig. 3. As mentioned earlier, a CNN incorporates numerous filters, each responsible for extracting distinct low-level or high-level features.

### Pooling layer
The result of a convolution operation is still a matrix, with the data number being the product of their height and width. The pooling layer aims to reduce the feature map's size, a process also known as sub-sampling. This not only enhances the algorithm's speed by reducing the number of parameters but also offers advantages in preventing overfitting and promoting network generalization. Pooling is executed by filters that operate differently from convolution filters, depending on their types. Various pooling operations exist, such as max pooling, average pooling, and global pooling. Maximum and average pooling are the most common, where the maximum or average value of a defined sub-matrix is transferred to the next layer, respectively. Importantly, it should be noted that the depth of the input matrix remains unchanged after pooling operations. An image before and after the max pooling operation is illustrated in Fig. 4.

### Fully connected layer
The subsequent layer following multiple convolution and pooling layers in a CNN architecture is the fully connected layer. Essentially, it constitutes a multi-layer neural network (*Rumelhart, Hinton & Williams, 1985*) that receives input from the last pooling

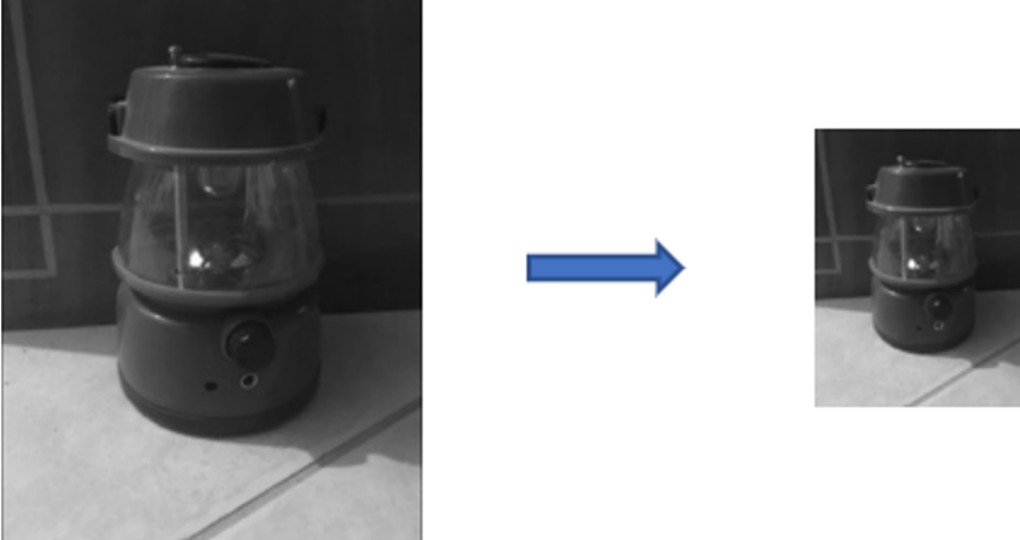

**Figure 4** **Input and output of a max-pooling operation.**

layer. However, the output of the pooling layer is not a vector and must be flattened (vectorized) before reaching this layer. While the flattening operation is sometimes considered an additional layer, it essentially involves rearranging the matrix into a vector. The number of hidden layers can vary in a fully connected layer. The transformation of information from the flattened input to the output follows the same principles as a multi-layer network. The output of each layer is computed using the equations below:

$$h_l^{in} = W_l h_{l-1}^{out} + B_l \tag{9}$$

$$y_l = \sigma(h_l^{in}) \tag{10}$$

where $h_{l-1}^{out}$ is the output vector of the previous hidden layer or the flattened vector, $W_l$ is the weight matrix between the current and previous layer, $B_l$ is the bias vector related with each neuron on the current layer, $h_l^{in}$ is the activation potential and $y_l$ is the output vector of the current layer. If the next layer is not the output layer, $y_l$ will serve as the $h_l^{out}$ for the next layer. $\sigma$ represents the activation function for each neuron, introducing nonlinearity to the network.

### Output layer
This layer is responsible for classification or decision-making and positioned at the end of the structure. In the case of a multi-class or multi-label classification problem, the number of neurons in the output layer corresponds to the number of classes/labels involved. Typically, for mutually exclusive classes, a softmax transfer function serves as the activation. However, in scenarios involving only two classes, such as in this study, a sigmoid transfer function is

preferred. This function produces output values between 0 and 1 for each output, allowing for the selection of a threshold value to convert the outputs into either 0 (negative) or 1 (positive).

## RESULTS AND DISCUSSIONS

The experimental procedure to validate the proposed method involves the following steps:

1. Generation of artificial images
2. Construction of a CNN structure
3. Training the CNN with artificial images
4. Fine-tuning the CNN with real images
5. Testing the CNN with unseen (untrained) real images
6. Comparing the obtained results with those classified by a conventional CNN using only real images

As outlined in 'Materials & Methods', artificial images are created employing Fresnel equations. These images primarily encompass two classes, representing normal and defected $MoS_2$ flakes on the $SiO_2/Si$ substrate. A normal flake denotes a uniformly grown single-layer, resulting in a triangular 2D $MoS_2$ flake, where mostly CVD grown structures are in triangular shape (*Wang et al., 2014*). On the other hand, defected $MoS_2$ flakes encompass structures with overlapping layers, single layers of non-triangular shapes (including grain boundaries) or irregular shapes, and multi-layer formations on the substrate.

It is important to note that the contrast values are calculated using the intensity values computed for two separate regions, as shown in Fig. 1. In region B, where no 2D material is present, only the refractive indices of $SiO_2$ and Si are used in the intensity calculation. However, when calculating intensity in region A for both normal and defective flakes, the layer containing $MoS_2$ is considered. Normal images are generated using the refractive indices of a single-layer $MoS_2$ with a thickness of $d_1 = 0.63$ nm (*Zhang et al., 2021*), along with the underlying $SiO_2$ and Si. The preferred thickness value for $SiO_2$ ($d_2$) is 300 nm because high contrast is achieved at this thickness (*Zhang et al., 2021*). The first subclass of defective images corresponds to overlapping flakes, where the reflection intensity value is calculated with the thickness value $2xd_1$. In the second defective case, reflection intensity values in different regions match those in the normal class, but the shapes are irregular. The last subclass of defective flakes consists of two or more layers where the growing material thickens in the middle of the flakes. In this case, the thickness of $MoS_2$ is represented by $n_l xd_1$ where $n_l$ is the number of layers. Refractive indices of $MoS_2$ vary depending on its thickness and are listed in Table 1 for three colors (*Hsu et al., 2019*; *Song et al., 2019*). Appropriate values are substituted into the intensity equations during artificial image generation.

$MoS_2$ flake shapes are randomly generated for various classes and positioned on an artificial image. The intensity value at each point on the image is computed using the previously described equations by substituting related refractive indices and thicknesses. It is important to emphasize that the intensity values are determined for three primary colors: red, green, and blue. These three RGB channels are then extracted and employed

**Table 1 Refractive indices.**

| $n$ | Red | Green | Blue |
| --- | --- | --- | --- |
| $n_{1layer}(MoS_2)$ | 4.3–1$i$ | 4.0–0.6$i$ | 4.3–1.3$i$ |
| $n_{2layer}(MoS_2)$ | 4.7–1.3$i$ | 5.1–1$i$ | 5.1–2.2$i$ |
| $n_{10layer}(MoS_2)$ | 4.3–0.7$i$ | 4.2–0.8$i$ | 4.5–2.5$i$ |
| $n_2$ (SiO$_2$) | 1.472 | 1.474 | 1.48 |
| $n_3$ (Si) | 3.8–0.002$i$ | 4.04–0.06$i$ | 4.69–0.09$i$ |

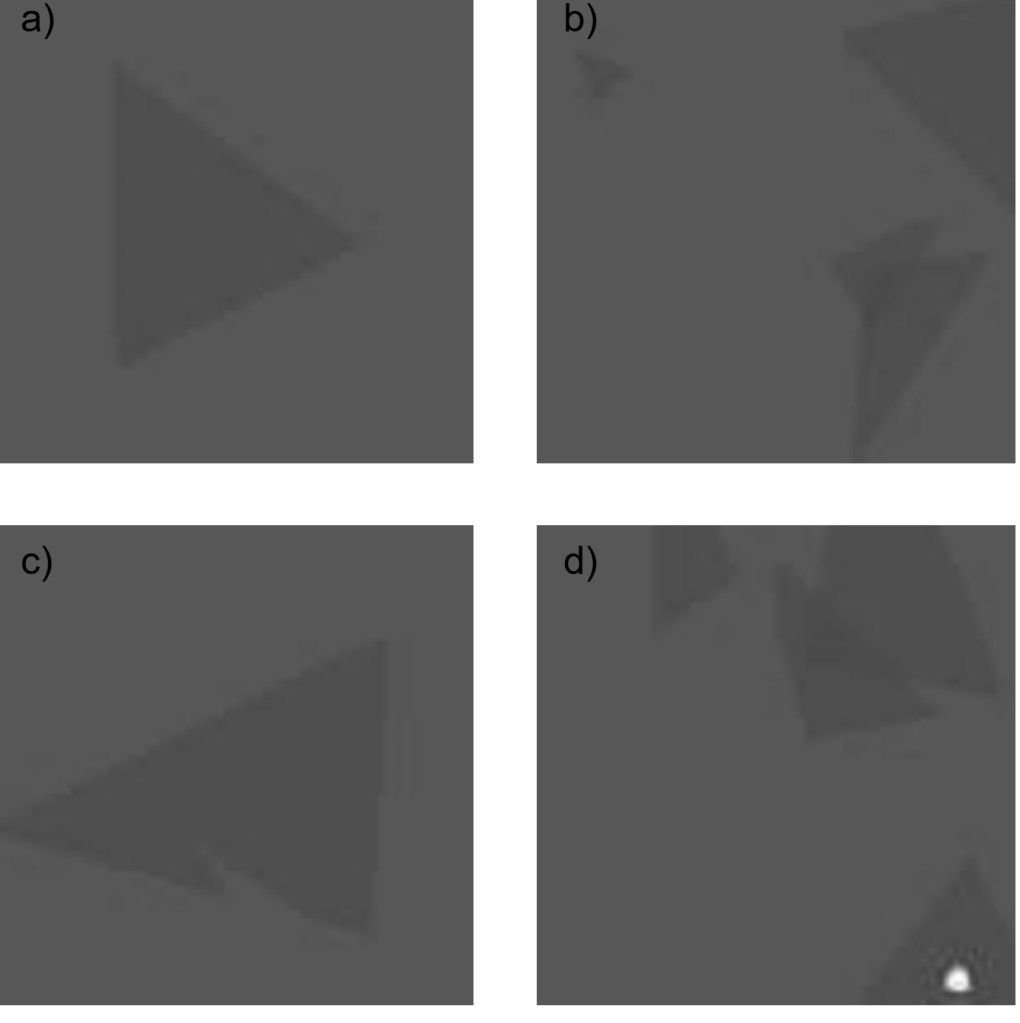

**Figure 5 Artificially generated 2D MoS$_2$ structures.** (A) Only normal flakes, (B) normal and overlapped flakes, (C) one-layer of non-triangular shape flake, and (D) overlapped, non-triangular shape and multi-layer formations on the substrate.

to generate a single artificial sample. Subsequently, the color images are transformed into grayscale images. Examples of artificially generated images are shown in Fig. 5.

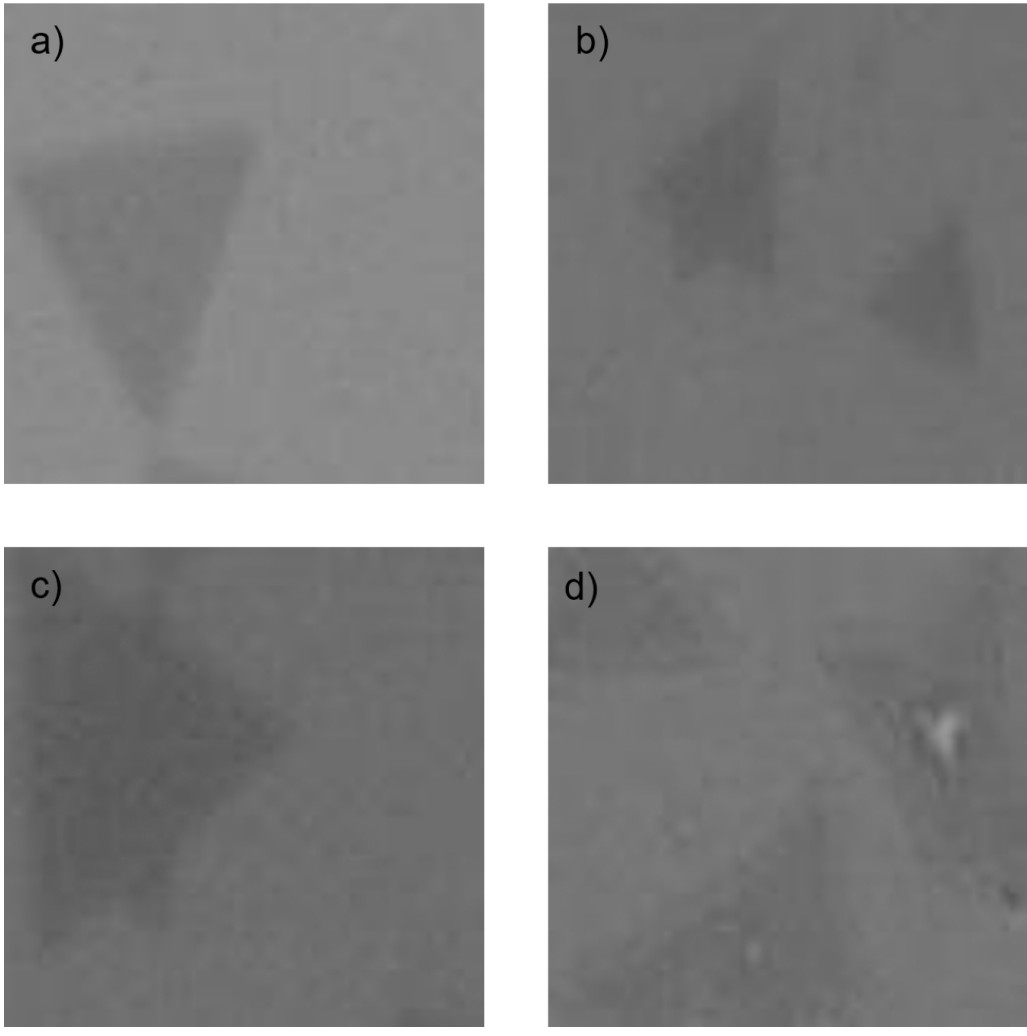

**Figure 6** **Experimentally obtained 2D MoS$_2$ structures.** (A) Only normal flakes, (B) normal and overlapped flakes, (C) one-layer of non-triangular shape flake, and (D) overlapped, non-triangular shape and multi-layer formations on the substrate.

On the other hand, real images of MoS$_2$ samples experimentally grown on the SiO$_2$/Si substrate using the CVD technique are illustrated in Fig. 6. These images were obtained from a brightfield microscope remain unprocessed and maintained in a neutral state. In supervised learning methods, labeled data is essential. While artificial data can be automatically labeled during generation, real images conventionally need to be manually labeled by an expert. Artificial images are used in the initial training phase, while real images are used during the fine-tuning process. Both real and artificial images are sized as matrices of $100 \times 100$.

The CNN model for pre-training phase is structured with five sets of convolution and pooling layers, followed by fully connected (dense) layers and a dropout layer before the output layer, as outlined in Table 2.

**Table 2  The CNN structure and number of parameters.**

| Layer | Output shape | Number of parameters |
|---|---|---|
| Convolution1 | $100 \times 100 \times 8$ | 80 |
| Max Pooling1 | $50 \times 50 \times 8$ | – |
| Convolution2 | $50 \times 50 \times 16$ | 1,168 |
| Max Pooling2 | $25 \times 25 \times 16$ | – |
| Convolution3 | $25 \times 25 \times 32$ | 4,640 |
| Max Pooling3 | $12 \times 12 \times 32$ | – |
| Convolution4 | $12 \times 12 \times 64$ | 18,496 |
| Max Pooling4 | $6 \times 6 \times 64$ | – |
| Convolution5 | $6 \times 6 \times 128$ | 73,856 |
| Max Pooling5 | $3 \times 3 \times 128$ | – |
| Flatten | 1,152 | – |
| Dense | 128 | 147,584 |
| Dense | 64 | 8,256 |
| Dropout | 64 | – |
| Dense | 1 | 9 |

**Table 3  Hyperparameters of the CNN algorithm.**

| Hyperparameter | Value |
|---|---|
| Learning rate | 0.0001 |
| Batch size | 32 |
| Number of epochs | 50 |

In the development of the CNN algorithm, a variety of hyperparameters were systematically tuned to enhance the model's performance. This tuning process involved a combination of empirical experiments and grid search, with the optimal results outlined in Table 3. It is crucial to acknowledge that the selection of hyperparameters may be contingent upon the specific characteristics of the dataset and the goals of the study.

Considering the general increase in validation accuracy after approximately 10 epochs during experiments, the maximum number of epochs was set to 50 to sufficiently capture evolving model dynamics. Following the completion of these epochs, the best-performing network was identified and isolated for subsequent use in the fine-tuning phase. It is noteworthy that the network's weights were initialized randomly, introducing variability at the commencement of the learning process. The Adam optimizer, acknowledged for its adeptness in managing adaptive learning rates, was employed as the optimization algorithm. Additionally, the binary cross-entropy loss function was adopted to quantify dissimilarity between predicted and actual class labels during the training process.

The number of filters in the convolutional layers is 8, 16, 32, 64, and 128, respectively. The size of each filter is $3 \times 3$ in the successive layers, where the first and second dimensions represent the height and width of the filters. The transfer (activation) functions in

convolutional layers are ReLU transfer functions given in Eq. (11).

$$y(x) = \max(0, x) \tag{11}$$

The first three pooling operations in the model are performed by dividing the matrices into $2 \times 2$ submatrices and selecting the maximum value of these.

After the extraction of meaningful and useful features from images through these layers, fully connected layers follow, taking flattened input from the last pooling layer. In this stage, every neuron in each hidden layer establishes full connections with both the preceding and succeeding layer neurons through their weight matrices. The output of each neuron is determined using the ReLU activation function, except for the output layer, where image classification as normal or defected using a sigmoid transfer function defined by the equation below.

$$y(x) = \frac{1}{1 + e^{-x}} \tag{12}$$

The fine-tuning phase follows the pre-training stage, typically utilizing the same layers and frozen weight parameters present in the network structure during the pre-training phase up to the flattening step. New layers replace the fully connected layers, and only their weights undergo training using real images. In this study, the structure identical to that employed in the pre-training phase outlined in Table 2 is adopted to fine-tuning CNN, with the exception of introducing an 8-output dense layer instead of the dropout layer. Furthermore, the weights associated with the last convolutional layer are not frozen but are subject to training during the fine-tuning process. The underlying concept is to enable subsequent convolutional layers in the CNN to capture more high-level features. Simultaneously, parameters recognizing such complex features in artificially generated images are removed from the CNN's memory, enhancing its ability to discern new images.

Finally, it is worth noting that the image classification task exploits two main strategies to counter the risk of overfitting. Initially, during the pre-training phase, a dropout layer was incorporated into the neural network architecture as a preventive measure against model overgeneralization, achieving this by stochastically deactivating neurons. Additionally, an early stopping mechanism was introduced, actively monitoring the model's performance on a separate validation set. When signs of overfitting emerge, as indicated by a degradation in performance on the validation set, the network parameters are kept as the optimal solution. These measures collectively contribute to the development of a robust and resilient model, proficient in addressing overfitting challenges throughout both the pre-training and subsequent fine-tuning phases. The optimal solution obtained from the pre-training phase is used for fine-tuning, and the optimal solution from the fine-tuning phase is employed to assess performance on test data.

To summarize, a CNN algorithm, enhanced with transfer learning and data augmentation, was developed for the classification of $MoS_2$ samples grown with the CVD technique as normal or defective, in this study. A normal sample should only contain normal flakes, while a defective sample must have at least one defective flake. For the sake of comparison, a classification was also performed with the same CNN structure and

hyperparameters that does not utilize transfer learning. For this purpose, the following two tasks were defined.

**Task0:** Classification of real images **without utilizing** transfer learning and artificially generated images. The dataset is divided into two classes, where class 0 encompasses defective flakes, and class 1 includes at least one normal flake.

**Task1:** Classification of real images **utilizing** transfer learning and artificial images. The dataset is divided into two classes, where class 0 encompasses defective flakes, and class 1 includes at least one normal flake.

Task 0 was conducted on laboratory-produced samples using the CNN structure employed in the fine-tuning phase. It is important to note that all parameters are unfrozen and initialized at the beginning. As mentioned before, the number of these samples is limited, and it may not yield satisfactory results for the training of a CNN. The total number of real data available is 290, with 145 normal and 145 defective samples. 60% of the real data was used for training, and the rest was used for validation and testing. For the second and main objective (Task1), the CNN was initially trained with artificially generated data using Fresnel equations (pre-training). A total of 5,000 artificial images were created, divided into training, validation, and test data sets at proportions of 80%, 10%, and 10%, respectively. Pre-training with artificial data was completed, and subsequently fine-tuning was performed using real data. It is important to bear in mind that, within the framework of the transfer learning method, during the fine-tuning process, certain layers at the beginning of the network structure have their parameters fixed (frozen), except for the fully connected layers, or occasionally some last convolution layers. In fine-tuning step, the same training, validation, and test sets used in the initial classification process (Task0) were employed to ensure a correct comparison.

The operational environment for the algorithm relies on a desktop computer running Windows 10 Enterprise as its operating system and implemented in Python 3.9. The computational power is provided by an Intel(R) Core(TM) i9-10920X CPU @ 3.50 GHz, with 12 cores and a clock speed of 3,504 MHz. The algorithm goes through a two-phase process; 1 epoch takes 1.871 s in the pre-training phase and 0.178 s in the fine-tuning phase. The total simulation time of both phases is 138 s on average. Analyzing the algorithm's fine-tuning and pre-training durations alongside the respective datasets reveals a high degree of computational efficiency and scalability. This is attributable to the nearly identical model architecture employed in both phases. Notably, with a 25-fold increase in the volume of data, the simulation time experiences only a 5-fold increment. This observation underscores the algorithm's efficiency and scalability, signifying that the computational demands grow at a rate significantly lower than the corresponding increase in dataset size. Such efficiency and scalability metrics affirm the model's robustness and resource-effective behavior, showcasing its adeptness in handling larger datasets without substantial computational burdens.

Various performance metrics, such as accuracy, precision, recall, and $F_1$ score (Eqs. (13)–(16)), are accessible for classification problems. Relying solely on one of them to assess the algorithm's performance may not be a prudent decision.

**Table 4  Performance measures of classification problems w.r.t the best accuracy measurement.**

| Problem | Accuracy | Precision | Recall | $F_1$ score |
|---------|----------|-----------|--------|-------------|
| Task0 | 78% | 81% | 72% | 76% |
| Task1 | 90% | 96% | 83% | 89% |

$$Accuracy = \frac{TP + TN}{TP + TN + FP + FN} \times 100 \tag{13}$$

$$Precision = \frac{TP}{TP + FP} \times 100 \tag{14}$$

$$Recall = \frac{TP}{TP + FN} \times 100 \tag{15}$$

$$F_1 Score = 2 \times \frac{Precision \times Recall}{Precision + Recall} \tag{16}$$

where *TP* is the number of true positives (correctly classified positives), *TN* is the number of true negatives (correctly classified negatives), *FP* is the misclassified positives and *FN* is the misclassified negatives. As mentioned before, positive and negative indicate existence and nonexistence of a normal flake, respectively.

Accuracy represents the percentage of correctly classified positive and negative samples among the total number of samples. However, accuracy alone may not provide sufficient insight, especially in the case of imbalanced data. Precision, the percentage of correct predictions within positive samples, and recall, the percentage of correctly classified positives within all positive predictions, offer a more nuanced evaluation. Precision and recall are trade-offs, and they are often considered together in the $F_1$ score, which represents the harmonic mean of these metrics.

The specified classification problems (with (Task1) and without (Task0) utilizing transfer learning) were addressed using the proposed CNN model. The training of the model begins with a randomly generated initial set of values. While it has the potential to reach the global minimum point, it often converges to local minimum points. Therefore, instead of a single simulation, the CNN algorithm was run 10 times for each of the two tasks, and the average of the results was recorded. The average accuracy metrics for Task0 and Task1 across 10 simulations are 68% and 85%, respectively. Therefore, a 17% improvement has been achieved with the proposed method. The best and worst results among 10 simulations for both tasks are presented in Tables 4 and 5, showing performance metrics: accuracy, precision, recall, and $F_1$ score.

The confusion matrix, which includes the classification results for a single experiment, is also provided in Fig. 7 to facilitate an assessment of the model's performance.

The results obtained for the classification of CVD-grown 2D MoS$_2$ images indicate lower performance when employing a standard CNN compared to the transfer learning-enhanced

**Table 5** Performance measures of classification problems w.r.t the worst accuracy measurement.

| Problem | Accuracy | Precision | Recall | $F_1$ score |
|---------|----------|-----------|--------|-------------|
| Task0 | 50% | 50% | 100% | 67% |
| Task1 | 81% | 78% | 86% | 82% |

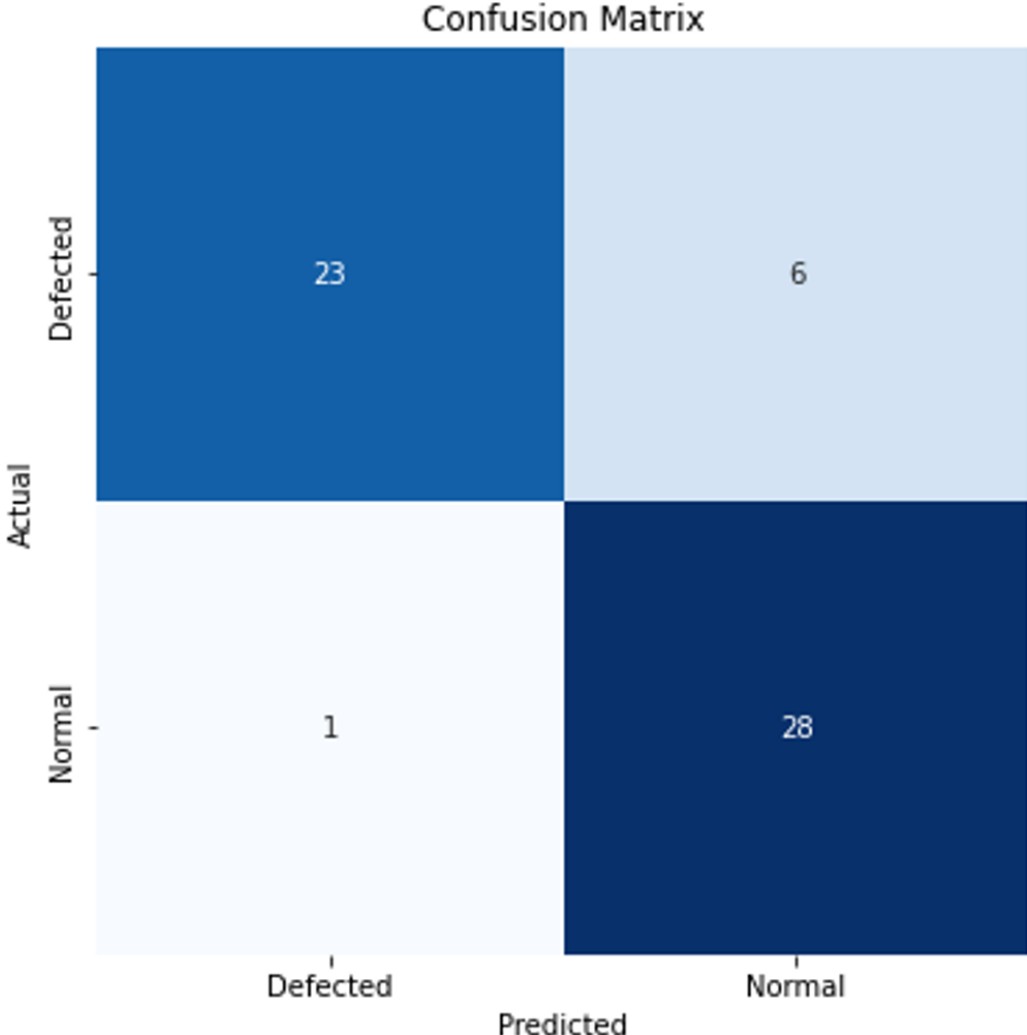

**Figure 7** Confusion matrix for a single experiment.

CNN. This outcome aligns with expectations, as machine learning methods typically exhibit suboptimal performance with limited data. However, a notable increase in accuracy is evident when utilizing the transfer learning method, with artificial images generated using Fresnel Equations. The tabulated results further demonstrate improvements in performance measures, ranging from a minimum of 11% to a maximum of 13% according to the best results of Task0 and Task1.

The proposed method effectively mitigates the challenge of limited data for CNN by employing a strategic combination of pre-training with Fresnel equations and fine-tuning with real images. The automated nature of the process enhances operational efficiency and minimizes manual intervention, underscoring its significant advantage for scalability. It is essential to acknowledge, however, that the model's performance can be further enhanced with an increased volume of real images. Additionally, the requirement for hyperparameter tuning necessitates careful attention to optimize the model's configuration. While our method provides an innovative solution to data scarcity and streamlines the process, the potential for performance improvement with additional real images warrants a thorough understanding of its applicability and considerations for refinement in future applications.

## CONCLUSIONS

The fabrication and analysis of 2D materials, such as CVD-grown $MoS_2$ structures, present significant challenges due to the intricate growth process, the necessity for specialized expertise, and the considerable time investment required. Deep learning methods, particularly convolutional neural networks (CNNs), provide a solution for analyzing and characterizing microscopic images of these 2D materials.

The efficient operation of CNN algorithms, similar to other machine learning problems, requires a substantial amount of data. The quantity and diversity of data play a crucial role in accurately completing the learning phase and preventing overfitting by avoiding neural network memorization. However, the production of these 2D materials involves extensive experiments, and labeling the images is a time-consuming process. Consequently, obtaining a sufficient amount of real data (labeled microscope images) to train a CNN is challenging. To address this challenge, a CNN with improved performance through the transfer learning method, is proposed, utilizing similar data. However, acquiring the mentioned similar data still poses a challenge for the problem presented in this article. This issue was overcome by generating images that resemble real data. A unique strategy utilizing artificial images generated through Fresnel equations was adopted. Virtual microscope images, containing 2D $MoS_2$ flakes, were created by considering intensity values based on the material properties in each region on a virtual sample. This approach not only mitigates data scarcity but also significantly enhances analysis accuracy. Further optimization of the CNN algorithm was achieved through meticulous fine-tuning with real images, contributing to the refinement of overall classification system. As a result of this innovative approach, significant improvements were observed in all measurement metrics. For the CNN trained with the existing limited data, the average accuracy value for binary classification was 68%. In contrast, when employing the transfer learning method and artificially created images using Fresnel equations, the same CNN demonstrated an average accuracy of 85%. This represents an average increase of approximately 17%. These findings underscore the potential of CNN algorithms in accelerating nanofabrication research processes, particularly in the development of devices with superior properties. While this study focuses on $MoS_2$, it is crucial to recognize the versatility of the methodology. The transfer learning-based CNN approach can seamlessly extend its applicability to other 2D materials by considering their specific parameters, such as refractive index and thickness.

This adaptability not only broadens the scope of the proposed method but also enhances its generalizability, facilitating advancements in the fabrication and analysis of diverse 2D materials.

However, it is important to recognize the potential for enhancing the model's performance with an increased number of real images. Besides, achieving optimal model configuration requires careful attention to hyperparameter tuning. In future research, exploring the generation of artificial images using alternative AI deep learning techniques alongside the utilization of Fresnel equations for pre-training holds potential for further enriching the diversity of the training dataset. Investigating methods such as generative adversarial networks (GANs) or variational autoencoders (VAEs) could contribute to a more comprehensive understanding of the model's robustness and adaptability in image classification tasks. This approach may open new possibilities for improving the performance and generalization of the CNN in diverse real-world scenarios.

Ethically, this study not only avoids negatively impacting human life, societal and social relationships but also has the potential to make a positive contribution to technological advancement.

## ACKNOWLEDGEMENTS

I would like to express my gratitude to the Micro Nano Devices and Systems (MIDAS) laboratory at Eskişehir Technical University for generously providing the optical microscopy images used in this research. Special thanks to Prof. Feridun Ay, Prof. Nihan Kosku Perkgoz, and their students, Mehmet Nacar and Arif Kayahan, for the acquisition of the optical microscopy images.

### Funding

This work was supported by the Eskisehir Technical University Scientific Research Projects Commission (No. 22ADP372 and 23ADP168). The funders had no role in study design, data collection and analysis, decision to publish, or preparation of the manuscript.

### Grant Disclosures

The following grant information was disclosed by the author:
The Eskisehir Technical University Scientific Research Projects Commission: 22ADP372, 23ADP168.

### Competing Interests

The authors declare there are no competing interests.

### Author Contributions

- Cahit Perkgoz conceived and designed the experiments, performed the experiments, analyzed the data, performed the computation work, prepared figures and/or tables, authored or reviewed drafts of the article, and approved the final draft.

## Data Availability

The real and artificially generated images, CNN python code without transfer learning, CNN python code for pretraining with artificial images, and the fine tuning CNN python code are available in Supplementary Files.

## Supplemental Information

Supplemental information for this article can be found online at http://dx.doi.org/10.7717/peerj-cs.1885#supplemental-information.

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
