# Peer review of "Identifying optical microscope images of CVD-grown two-dimensional MoS2 by convolutional neural networks and transfer learning"

_PeerJ Computer Science, doi:10.7717/peerj-cs.1885_

## Round 0.1 · original submission · Major Revisions

Dear authors,

Thank you for submitting your article. Reviewers have now commented on your article and suggest major revisions. When submitting the revised version of your article, it will be better to address the following:

1- Keywords should be written in alphabetical order.

2- The research gaps and contributions should be clearly summarized in the introduction section. Please evaluate how your study is different from others in the related work section.

3- Please include future research directions.

4- The paper lacks the running environment, including software and hardware. The analysis and configurations of experiments should be presented in detail for reproducibility. It is convenient for other researchers to redo your experiments and this makes your work easy acceptance. A table with parameter settings for experimental results and analysis should be included in order to clearly describe them.

5- The authors should clarify the pros and cons of the methods. What are the limitation(s) methodology(ies) adopted in this work? Please indicate practical advantages, and discuss research limitations.

6- Explanation of the equations should be checked. All variables should be written in italics as in the equations.

7- Reviewers have requested that you cite specific references. You may add them if you believe they are especially relevant. However, I do not expect you to include these citations, and if you do not include them, this will not influence my decision.

Best wishes,

**Language Note:** PeerJ staff have identified that the English language needs to be improved. When you prepare your next revision, please either (i) have a colleague who is proficient in English and familiar with the subject matter review your manuscript, or (ii) contact a professional editing service to review your manuscript. PeerJ can provide language editing services - you can contact us at copyediting@peerj.com for pricing (be sure to provide your manuscript number and title). – PeerJ Staff

Reviewer 1 ·

Basic reporting

The study aims to use artificial intelligence, specifically Convolutional Neural Networks (CNNs), for analyzing 2D structures grown through the CVD technique, focusing on their suitability for electronic device applications. The methodology involves generating artificial images using Fresnel equations to augment the dataset for CNN training, addressing data scarcity issues. However, the study have some areas that needs improvement, hence the authors needs to revise their article using these comments to improve the quality of the study.
1. The study relies heavily on artificially generated images using Fresnel equations. While this addresses the challenge of limited real data, there's a risk that these artificial images may not fully capture the complexity and variability found in actual microscope images.
2. The study's focus on MoS2 structures may limit its applicability to other 2D materials. While the authors suggest that the methodology can be extended to other materials, this claim needs validation through further experiments.
3. The validation of the CNN model primarily utilizes a limited set of real images, which might not be sufficient to comprehensively assess the model's robustness and accuracy in diverse real-world scenarios.
4. The study mentions using artificial images to prevent overfitting, but it doesn't explicitly discuss the steps taken to ensure the CNN model is not overfitting on these images.
5. Real images require manual labeling, which is time-consuming and prone to human error. The study could explore automated or semi-automated labeling techniques to enhance efficiency and accuracy.
6. The transfer learning approach is promising, but the study could benefit from a more detailed exploration of how the transfer learning model was adapted and fine-tuned using real data.
7. The computational efficiency and scalability of the proposed CNN model, especially when handling large datasets or extended to other 2D materials, are not discussed.
8. More detailed information on the CNN architecture and training parameters would aid in replicability and independent verification of the study's results.
9. The study lacks a comprehensive comparison with other image classification methods, both conventional and AI-based, which could provide a clearer context for the CNN model's performance.
10. The study doesn’t discuss the ethical implications and practical challenges of implementing this technology in real-world industrial settings.
11. There is a need for significant improvement in the general quality of the English language and its presented method. A substantial quantity of typographical errors and grammatical issues were observed. To ensure the linguistic accuracy and subject familiarity of your paper, it is advisable to engage the assistance of a colleague proficient in English and knowledgeable in the issue or seek the services of a professional editing firm.
12. We are in December 2023, and I could not see any 2022 and 2023 citations. Citing recent literature has several advantages for both authors and journals. It can help authors establish their credibility, demonstrate their research's relevance, and help avoid plagiarism. In the same way, it assists journals in increasing their visibility, improving their reputation, increasing their citation rates, and meeting reader expectations. For this reason, I have suggested some recent literature from 2024, 2023 and 2022 relating to the study that you must cite and reference in your article.

a. Lu, G., Duan, L., Meng, S., Cai, P., Ding, S.,... Wang, X. (2023). Development of a colorimetric and turn-on fluorescent probe with large Stokes shift for H2S detection and its multiple applications in environmental, food analysis and biological imaging. Dyes and Pigments, 220, 111687. doi: https://doi.org/10.1016/j.dyepig.2023.111687
b. Lu, G., Yu, S., Duan, L., Meng, S., Ding, S.,... Dong, T. (2024). New 1,8-naphthalimide-based colorimetric fluorescent probe for specific detection of hydrazine and its multi-functional applications. Spectrochimica Acta Part A: Molecular and Biomolecular Spectroscopy, 305, 123450. doi: https://doi.org/10.1016/j.saa.2023.123450
c. Zhang, R., Li, L., Zhang, Q., Zhang, J., Xu, L., Zhang, B.,... Wang, B. (2023). Differential Feature Awareness Network within Antagonistic Learning for Infrared-Visible Object Detection. IEEE Transactions on Circuits and Systems for Video Technology. doi: 10.1109/TCSVT.2023.3289142
d. Dong, W., Zhao, J., Qu, J., Xiao, S., Li, N., Hou, S.,... Li, Y. (2023). Abundance Matrix Correlation Analysis Network Based on Hierarchical Multihead Self-Cross-Hybrid Attention for Hyperspectral Change Detection. IEEE Transactions on Geoscience and Remote Sensing, 61. doi: 10.1109/TGRS.2023.3235401
e. Dong, W., Yang, Y., Qu, J., Xiao, S., & Li, Y. (2023). Local Information-Enhanced Graph-Transformer for Hyperspectral Image Change Detection With Limited Training Samples. IEEE Transactions on Geoscience and Remote Sensing, 61. doi: 10.1109/TGRS.2023.3269892
f. Li, S., Chen, J., Peng, W., Shi, X., & Bu, W. (2023). A vehicle detection method based on disparity segmentation. Multimedia Tools and Applications, 82(13), 19643-19655. doi: 10.1007/s11042-023-14360-x
g. Yin, Y., Guo, Y., Su, Q., & Wang, Z. (2022). Task Allocation of Multiple Unmanned Aerial Vehicles Based on Deep Transfer Reinforcement Learning. Drones, 6(8), 215. doi: 10.3390/drones6080215
h. Qi, M., Cui, S., Chang, X., Xu, Y., Meng, H., Wang, Y.,... Arif, M. (2022). Multi-region Nonuniform Brightness Correction Algorithm Based on L-Channel Gamma Transform. Security and communication networks, 2022. doi: 10.1155/2022/2675950

Experimental design

• While using Fresnel equations to generate artificial images for CNN training is innovative, it raises concerns about the model's ability to generalize to real-world scenarios. This dependence on synthetic data might not fully capture the complexity and variability found in actual MoS2 samples.
• The study's approach to fine-tuning the CNN with a small set of real images could limit the model's robustness. The real-world applicability and effectiveness of the CNN might be constrained if it is not exposed to a sufficiently diverse and representative dataset.
• The study might benefit from testing the CNN under various conditions, such as lighting or imaging settings. This would help assess the model's adaptability and performance consistency across different practical scenarios.

Validity of the findings

• While the study suggests that the methodology could be extended to other 2D materials, this claim needs further validation. Testing the approach with different materials, each having unique optical properties, is essential to substantiate the generalizability of the findings.
• The study could strengthen its findings by quantitatively validating the artificial images against a more extensive set of real images. This would ensure that the synthetic data accurately represents the range of features and variations in actual MoS2 samples.
• While significant, the improvement in classification accuracy would benefit from a deeper analysis. Understanding the model's performance regarding false positives and negatives, especially in a real-world setting, is crucial for validating its practical utility.
• Additional experiments, possibly with independent datasets or in collaboration with other research groups, would help verify the findings' replicability and robustness. This would add credibility to the study's conclusions and support broader adoption of the proposed methodology.

Additional comments

• Could the authors elaborate on their method for tuning the optimal hyperparameters of all models? A clear description is needed.
• What strategies were employed to address the issues of overfitting and the limited size of the dataset?
• The study's limitations and suggestions for potential future research should be articulated.
• It would be beneficial if the source codes were made available to facilitate study replication.
• A comparison of this study with current state-of-the-art systems is necessary. The authors should explain how their study outperforms or falls short of these existing systems.

Reviewer 2 ·

Basic reporting

The aim of this study is to address the challenges in the production and analysis of two-dimensional materials, particularly MoS2, for electronic device applications. The core of the research is the development of a transfer learning-based deep Convolutional Neural Network (CNN) to categorize the uniformity of CVD-grown MoS2 flakes and identify defects affecting electronic properties.

Experimental design

The application of transfer learning and CNNs to improve the classification of 2D materials based on their quality is innovative. The use of Fresnel equations to generate artificial images for pre-training is a thoughtful approach to overcome the scarcity of labeled data.

Validity of the findings

The method significantly improved classification accuracy from 68% to 85% compared to ordinary CNNs. This demonstrates the potential of the approach in real-world applications and could be a substantial contribution to the field.

Additional comments

1. While the results are promising, the robustness of the model could be further validated by testing with a broader range of datasets from different sources to ensure its adaptability and accuracy in various scenarios.
2. The study could benefit from a more extensive comparative analysis with other state-of-the-art methods or different deep learning approaches.
3. The author should add a literature related work section before the material and methods section.
4. It is expected that the author should highlight the contribution of the study in the concluding paragraph of the introduction section.
5. It was also discovered that did author did not use some latest or reference some latest work. To enhance the quality of the work, the suggest references are recommended for citation in the manuscript. the following references can be use to build a related work as suggested in comment 3.

1. Dong, Y., Xu, B., Liao, T., Yin, C., & Tan, Z. (2023). Application of Local-Feature-Based 3-D Point
Cloud Stitching Method of Low-Overlap Point Cloud to Aero-Engine Blade Measurement. IEEE
Transactions on Instrumentation and Measurement, 72. doi: 10.1109/TIM.2023.3309384
2. Shi, Y., Xi, J., Hu, D., Cai, Z., & Xu, K. (2023). RayMVSNet++: Learning Ray-Based 1D Implicit Fields
for Accurate Multi-View Stereo. IEEE Transactions on Pattern Analysis and Machine Intelligence,
45(11), 13666-13682. doi: 10.1109/TPAMI.2023.3296163
3. Fu, C., Yuan, H., Xu, H., Zhang, H., & Shen, L. (2023). TMSO-Net: Texture adaptive multi-scale
observation for light field image depth estimation. Journal of Visual Communication and Image
Representation, 90, 103731. doi: https://doi.org/10.1016/j.jvcir.2022.103731
4. Zhao, C., Cheung, C. F., & Xu, P. (2020). High-efficiency sub-microscale uncertainty measurement
method using pattern recognition. ISA Transactions, 101, 503-514. doi:
https://doi.org/10.1016/j.isatra.2020.01.038
5. Ye, X., Wang, J., Qiu, W., Chen, Y., & Shen, L. (2023). EXCESSIVE GLIOSIS AFTER VITRECTOMY FOR
THE HIGHLY MYOPIC MACULAR HOLE: A Spectral Domain Optical Coherence Tomography Study.
RETINA, 43(2). doi: 10.1097/IAE.0000000000003657
6. Qian, J., Cao, Y., Bi, Y., Wu, H., Liu, Y., Chen, Q.,... Zuo, C. (2023). Structured illumination
microscopy based on principal component analysis. eLight, 3(1), 4. doi: 10.1186/s43593-022-
00035-x
7. Li, J., Zhou, N., Sun, J., Zhou, S., Bai, Z., Lu, L.,... Zuo, C. (2022). Transport of intensity diffraction
tomography with non-interferometric synthetic aperture for three-dimensional label-free
microscopy. Light: Science & Applications, 11(1), 154. doi: 10.1038/s41377-022-00815-7

6. The author is require to thoroughly proof read the manuscript for likely grammar error.
7. The author is expected to give the limitation of the study in the concluding section.

---

## Round 0.2 · accepted · Accept

Dear author,

Thank you for the revision. The paper seems to be improved in the opinion of the reviewers. The paper is now ready for publication.

Best wishes,

Reviewer 1 ·

Basic reporting

I am delighted to convey that following a meticulous review, your manuscript has undergone all requisite corrections successfully. The revisions and updates you have incorporated have markedly elevated the overall quality of the paper. Consequently, I am recommending your article for publication. Your diligence and efforts in addressing the previous concerns are commendable, and the manuscript is now poised to make a noteworthy contribution to the field.

Experimental design

No comment

Validity of the findings

No comment

Additional comments

No comment

Reviewer 2 ·

Basic reporting

All the comments raised in this section has been attended to by the authors.

Experimental design

All the comments raised in this section has been attended to by the authors.

Validity of the findings

All the comments raised in this section has been attended to by the authors.

Additional comments

All the comments raised in this section has been attended to by the authors.